# The Hexane Extract of *Citrus sphaerocarpa* Ameliorates Visceral Adiposity by Regulating the PI3K/AKT/FoxO1 and AMPK/ACC Signaling Pathways in High-Fat-Diet-Induced Obese Mice

**DOI:** 10.3390/molecules28248026

**Published:** 2023-12-09

**Authors:** Liqing Zang, Kazuhiro Kagotani, Takuya Hayakawa, Takehiko Tsuji, Katsuzumi Okumura, Yasuhito Shimada, Norihiro Nishimura

**Affiliations:** 1Graduate School of Regional Innovation Studies, Mie University, Tsu 514-8507, Mie, Japan; nishimura.norihiro@mie-u.ac.jp; 2Mie University Zebrafish Drug Screening Center, Mie University, Tsu 514-8507, Mie, Japan; k.kagotani@tsuji-seiyu.co.jp (K.K.); shimada.yasuhito@mie-u.ac.jp (Y.S.); 3Tsuji Health & Beauty Science Laboratory, Mie University, Tsu 514-8507, Mie, Japan; t.hayakawa@tsuji-seiyu.co.jp (T.H.); ta.tsuji@tsuji-seiyu.co.jp (T.T.); 4Tsuji Oil Mills Co., Ltd., Matsusaka 515-0053, Mie, Japan; 5Department of Life Sciences, Graduate School of Bioresources, Mie University, Tsu 514-8507, Mie, Japan; katsu@bio.mie-u.ac.jp; 6Department of Integrative Pharmacology, Mie University Graduate School of Medicine, Mie University, Tsu 514-8507, Mie, Japan; 7Department of Bioinformatics, Mie University Advanced Science Research Promotion Centre, Tsu 514-8507, Mie, Japan

**Keywords:** obesity, citrus fruit, hexane extract, visceral adipose tissue, zebrafish, HFD-induced obese mice

## Abstract

Obesity is an emerging global health issue with an increasing risk of disease linked to lifestyle choices. Previously, we reported that the hexane extract of *Citrus sphaerocarpa* (CSHE) suppressed lipid accumulation in differentiated 3T3-L1 adipocytes. In this study, we conducted in vivo experiments to assess whether CSHE suppressed obesity in zebrafish and mouse models. We administered 10 and 20 μg/mL CSHE to obese zebrafish juveniles. CSHE significantly inhibited visceral fat accumulation compared to untreated obese fish. Moreover, the oral administration (100 μg/g body weight/day) of CSHE to high-fat-diet-induced obese mice significantly reduced their body weight, visceral fat volume, and hepatic lipid accumulation. The expression analyses of key regulatory genes involved in lipid metabolism revealed that CSHE upregulated the mRNA expression of lipolysis-related genes in the mouse liver (*Ppar*α and *Acox1*) and downregulated lipogenesis-related gene (*Fasn*) expression in epididymal white adipose tissue (eWAT). Fluorescence immunostaining demonstrated the CSHE-mediated enhanced phosphorylation of AKT, AMPK, ACC, and FoxO1, which are crucial factors regulating adipogenesis. CSHE-treated differentiated 3T3L1 adipocytes also exhibited an increased phosphorylation of ACC. Therefore, we propose that CSHE suppresses adipogenesis and enhances lipolysis by regulating the PI3K/AKT/FoxO1 and AMPK/ACC signaling pathways. These findings suggested that CSHE is a promising novel preventive and therapeutic agent for managing obesity.

## 1. Introduction

Obesity is caused by abnormal or excessive adipose tissue storage due to an imbalance between calorie intake and consumption [1]. According to the newest factsheet published by the WHO, over 1.9 billion adults were overweight (25 ≤ Body mass index (BMI) < 30) and over 650 million were obese (BMI ≥ 30) in 2016 [2]. Hypertension, cardiovascular diseases, diabetes, musculoskeletal disorders, osteoarthritis, and cancer are common in overweight and obese individuals. Therefore, obesity has been recognized as a disease [3]. Current treatments for obesity mainly focus on lifestyle management, such as a restricted calorie intake, physical activity, and medical or surgical therapy. However, lifestyle modifications alone are often difficult to maintain and have limited efficacy. Pharmacotherapy is associated with undesirable side effects, including headaches, dizziness, insomnia, and gastrointestinal effects [4]. Therefore, the use of natural products (NPs) and NP-derived bioactive compounds that counteract obesity with minimal side effects has attracted considerable attention [5].

Several NPs derived from plants, animals, and bacteria have been reported to ameliorate obesity and its related metabolic diseases [6,7,8]. Citrus fruits are well-known bioactive materials containing vitamins, phenolic acids, and flavonoids. Polymethoxyflavones (PMFs) are primarily present in citrus peels. PMFs include glycosides (hesperidin and naringin) and O-methylated flavone aglycones (nobiletin and tangeretin) [9]. PMFs play many beneficial roles in human health due to their anti-inflammatory, antioxidant, and anti-cancer properties [10,11,12,13]. Cell and animal studies have demonstrated that citrus-derived PMFs are important for preventing obesity, type 2 diabetes, and other metabolic disorders [14]. For instance, citrus-derived PMFs inhibit apoB secretion and regulate the lipid synthesis in HepG2 cells [15,16]. Aged citrus peel (Chenpi) extract suppressed the body weight gain and adipogenesis in obese mice [17]. PMFs derived from *Citrus aurantium* L. prevented high-fat-diet (HFD)-induced obesity in mice by promoting adaptive thermogenesis in adipose tissues [18]. The PMFs-associated mechanism underlying the suppressed adipogenesis promoted protein kinase A (PKA) signaling, which included the upregulated phosphorylation of protein kinase A catalytic subunit α (PKACα), adenosine monophosphate-activated protein kinase (AMPK), and acetyl-CoA carboxylase (ACC) in 3T3-L1 cells [19]. Moreover, Kim et al. demonstrated that *C. aurantium* flavonoids inhibit adipogenesis via suppressing the serine phosphorylation of protein kinase B (AKT), which responds to phosphoinositide 3-kinase (PI3K)-regulated lipid accumulation [20]. Citrus PMFs are important bioactive compounds suitable for the treatment of obesity.

*Citrus sphaerocarpa* (Japanese name: “Kabosu”) is a popular sour citrus used for vinegar, seasonings, jams, marmalades, and juice production in Japan. Due to their unique tastes, essential oils derived from citrus peels find extensive application in the food, beverage, pharmaceutical, and cosmetic sectors. Although the extracts of several citrus fruit peels exhibit anti-obesity activity, the effects of *C. sphaerocarpa* supplementation on obesity have rarely been reported. Recently, we demonstrated that the hexane extract of *C. sphaerocarpa* (CSHE) suppressed lipid accumulation in differentiated 3T3-L1 adipocytes [21]. RNA sequencing analysis revealed that this effect might be associated with the activation of β-oxidation through the PI3K/AKT and PKA/AMPK signaling pathways. However, further studies, including in vivo experiments, are required to demonstrate the preventive and therapeutic potential of CSHE against obesity.

The zebrafish (*Danio rerio*) is a well-established model organism for human disease studies [22,23]. We have developed zebrafish obesity models using adult and juvenile zebrafish [24,25] and identified numerous natural products exhibiting anti-obesity properties [26,27,28]. Here, we evaluated the anti-obesity effects of CSHE using obese zebrafish and mice. Alterations in the genes and proteins associated with lipid metabolism were evaluated in the mouse liver and epididymal white adipose tissue (eWAT) to elucidate the underlying molecular mechanisms, which were further validated using differentiated 3T3-L1 adipocytes.

## 2. Results

### 2.1. CSHE Decreases the Visceral Adipose Tissue (VAT) Accumulation in Obese Juvenile Zebrafish

CSHE mainly contains low-polarity and highly volatile chemicals, including D-limonene, β-myrcene, and γ-terpinene, which collectively make up over 85% of its chemical total content [21]. Nevertheless, these compounds did not play a role in inhibiting the lipid accumulation in differentiated 3T3-L1 cells [21]. Total ion chromatography and LC/MS analyses detected two hydroxylated PMFs, hydroxylated hexamethoxyflavone and hydroxylated tetramethoxyflavone, which are potential candidates contributing to the anti-obesity effects of CSHE (Appendix A).

To confirm the effect of CSHE on obesity, the zebrafish obesogenic test (ZOT) was performed on juvenile zebrafish. Using Nile Red staining to visualize neutral lipid droplets within live cells, we detected and quantified VAT accumulation in juvenile zebrafish. Lipids were detected at the posterior end of the digest system (Figure 1b). Moreover, the 20 μg/mL CSHE (CSHE20) treatment for one day significantly decreased the VAT volume compared to the control group treated with a HFD (−44%, *p* < 0.01; Figure 1c); however, the lower dose (10 μg/mL, CSHE10) induced a 17% decrease with no statistical significance. We performed a long-term experiment to evaluate the anti-obesity effects of CSHE10 (Figure 1d). Juvenile zebrafish were treated with CSHE10 for four days, and Nile Red staining was performed on days 1, 2, and 4. Nile Red fluorescence images revealed no difference in the VAT accumulation (Figure 1e) between the control and CSHE10 groups on day 2 (Figure 1f); however, CSHE10 induced a significant reduction in VAT volume after two more days of treatment (−78%, *p* < 0.05; Figure 1f).

### 2.2. CSHE Reduces the VAT and Hepatic Lipid Accumulation in HFD-Induced Obese Mice

To confirm whether the preventive effect of CSHE on VAT accumulation was common in mammals, CSHE was orally administered to HFD-treated mice (0.1% emulsion in drinking water, approximately 100 μg/g body weight/day). The body weight was significantly (*p* < 0.01) higher in HFD-fed control mice than in normal diet (ND)-fed mice at week 1 (Figure 2a). However, significant decreases (*p* < 0.05) were detected in the CSHE-treated mice compared to the HFD-induced obese control mice at all time points except at week 2. After 6 weeks of treatment, CSHE significantly suppressed the body weight of mice compared to the HFD group (31.5 ± 1.9 g vs. 34.9 ± 2.5 g, *p* < 0.05). There was no significant difference in the average food intake among the four groups (Figure 2b), indicating that CSHE did not suppress appetite. After the HFD treatment, fasting blood glucose, plasma triglyceride, and plasma total cholesterol levels did not differ significantly between the control and CSHE-treated groups (Figure 2c–e). A 3D-microcomputed tomography (3D-micro-CT) analysis demonstrated a significantly higher (*p* < 0.001) VAT volume in the HFD group (Figure 2f). However, CSHE intake suppressed the VAT accumulation compared to that in the HFD group (2085 ± 597 cm^3^ [HFD-CSHE] vs. 2662 ± 395 cm^3^ [HFD-control]) (*p* < 0.05). Representative 3D-micro-CT images (Figure 2g) revealed the decreased VAT volume in the CSHE-administered groups. Moreover, the CSHE-treated mice showed reduced hepatic lipid accumulation compared to the HFD group (Figure 2h). Image quantification using Oil Red O-positive tissue showed that CSHE significantly improved hepatic fatty deposition (−76%, *p* < 0.05; Figure 2i).

### 2.3. CSHE Prevents Lipid Accumulation via Modulating Lipid Metabolism Genes in the Liver and eWAT

Furthermore, relative gene expression profiles were analyzed to investigate whether CSHE regulates lipid metabolism, such as lipogenesis or lipolysis (β-oxidation). In the liver tissues, genes related to lipogenesis, such as peroxisome proliferator-activated receptor gamma (*Pparg*), sterol regulatory element binding transcription factor 1 (*Srebf1*), and CCAAT/enhancer binding protein (C/EBP) alpha and beta (*Cebpa* and *Cebpb*), were not significantly changed in the HFD–CSHE group compared to those in HFD control group. In HFD-induced obese mice, their fatty acid synthase (*Fasn*) levels increased dramatically after CSHE administration (*p* < 0.05; Figure 3a). In contrast, lipolysis and fatty acid β-oxidation-related genes, such as peroxisome proliferator-activated receptor alpha (*Ppara*) and its target genes, acyl-coenzyme A oxidase 1 and palmitoyl (*Acox1*), were significantly upregulated in CSHE-treated obese mice compared to in HFD control mice (*p* < 0.05, Figure 3b). The expression of peroxisome proliferator-activated receptor gamma, coactivator 1 alpha (*Ppargc1a*), carnitine palmitoyltransferase 1a, and liver (*Cpt1a*) did not vary between the HFD–CSHE and HFD groups.

In eWAT, CSHE significantly inhibited the mRNA expression of *Fasn* compared to in HFD-treated mice (*p* < 0.05; Figure 3c). Moreover, *Cebpa* and *Cebpb* were upregulated in CSHE-treated mice (*p* < 0.05, Figure 3c). The expression levels of lipolysis-related genes (*Ppara*) tended to increase in CSHE-treated mice compared to those in HFD-treated mice (*p* = 0.08; Figure 3d). The CSHE treatment significantly upregulated *Ppargc1a*, a gene playing a crucial regulatory role in energy metabolism (*p* < 0.05; Figure 3d). These findings indicate that the CSHE treatment promoted fatty acid β-oxidation-related genes, potentially contributing to the prevention of lipid accumulation in the liver and visceral adiposity.

### 2.4. CSHE Modulates AKT/AMPK and Forkhead Box Protein O1 (FoxO1)/ACC Activity in the Liver and eWAT

Using sections prepared from the liver and eWAT of the tested mice, we performed fluorescence immunostaining to investigate the PI3K/AKT and AMPK signaling pathway-related proteins and their phosphorylation levels. CSHE enhanced the phosphorylation of AKT, a representative protein kinase, in liver tissues (*p* < 0.05; Figure 4). Furthermore, CSHE enhanced AMPK phosphorylation in the liver tissues of obese mice (*p* = 0.098, Figure 5). The phosphorylation of FoxO1, which is controlled by AKT, increased in the eWAT after the CSHE administration (*p* < 0.05; Figure 6). In eWAT, the phosphorylation of ACC, which acts downstream of AMPK, demonstrated more activity in the HFD–CSHE group than in the HFD group (*p* < 0.05; Figure 7).

### 2.5. CSHE Treatment Activates ACC in Differentiated 3T3-L1 Adipocytes

Considering the high ACC phosphorylation detected in the eWAT of HFD mice treated with CSHE, we analyzed the ACC phosphorylation levels in drug-induced differentiated 3T3-L1 adipocytes treated with CSHE. Although the ACC expression levels were equivalent between the adipocytes treated with or without CSHE, ACC phosphorylation was reduced in differentiated adipocytes without CSHE (Figure 8a–c). However, ACC phosphorylation significantly increased (*p* < 0.001; Figure 8d), and a high rate of ACC phosphorylation was detected in 3T3-L1 cells after the CSHE treatment (*p* < 0.001; Figure 8e). The impact of CSHE on differentiated 3T3-L1 adipocytes mirrored the effects observed in the eWAT of CSHE-treated HFD mice.

## 3. Discussion

Obesity is associated with adipocyte differentiation and lipid accumulation. Adipocytes are promising targets for anti-obesity interventions, and the regulation of adipogenesis is believed to be influenced by various plant extracts [29]. Previously, CSHE was reported to suppress the lipid accumulation (adipogenesis) in 3T3-L1 adipocytes [21]; however, further research is needed to determine its anti-obesity effects and underlying molecular mechanisms. Zebrafish are a useful tool for evaluating anti-obesity products [24,25,26]. Therefore, in this study, we used juvenile obese zebrafish for our initial evaluation, considering their high efficacy and the short experimental period. One-day exposure to CSHE (20 μg/mL) significantly suppressed the visceral lipid accumulation in obese zebrafish juveniles; however, its low dose (10 μg/mL) showed no significant effect. A similar treatment using a low dose of CSHE for three days reduced lipid accumulation, indicating the potential anti-obesity effects of CSHE, even at a small dose (Figure 1). The anti-obesity effect of CSHE was further confirmed in a mammalian mouse model. The results were comparable to those of cultured cell- and zebrafish-based experiments (Figure 2), providing a new candidate for anti-obesity drug discovery.

Lipogenesis refers to synthesizing fatty acids and triglycerides from acetyl-CoA and malonyl-CoA, followed by their storage in cells such as adipocytes or hepatocytes [30]. An excessive caloric intake stimulates lipogenesis, leading to fat accumulation. Several key regulators, including PPARγ, FASN, sterol regulatory element binding protein (SREBP), and C/EBPs, are involved in lipogenesis. PPARγ is the key regulator of adipogenesis, associated with adipocyte differentiation and lipid storage [31]. FASN, a vital enzyme in controlling de novo lipogenesis, catalyzes the synthesis of long-chain fatty acids from malonyl-CoA [32]. PPARγ regulates the generation of malonyl-CoA from acetyl-CoA [33]. SREBP1 activates PPARγ, facilitates lipogenesis, and subsequently increases adipocyte fatty acid uptake and lipid accumulation [34]. Like PPAR, CEBPs play important roles in adipocyte differentiation and are associated with fatty acid metabolism [35]. The regulation of these lipogenic factors can reduce lipid synthesis and accumulation; hence, these factors are considered therapeutic molecular targets for obesity [36]. In this study, CSHE reduced the gene expression of *Fasn* in the eWAT of obese mice (Figure 3). Compared to the downregulatory effect of green tea extract on the *cebpa* expression level in overfed zebrafish [37], CSHE significantly upregulated *Cebpa* and *Cebpb* in the eWAT tissue of HFD-treated mice. This may promote adipocyte differentiation and regulate fatty acid uptake, thereby suppressing fat accumulation. Additionally, the upregulation of *Fasn* in the livers of CSHE-treated obese mice indicated enhanced fatty acid synthesis. This change does not imply a simple increase in fat accumulation and shows the possible correlation of other mechanisms with the suppressed lipid accumulation in the liver. These results indicated that the CSHE treatment may improve intracellular fatty acid metabolism by inhibiting lipogenesis, particularly in eWAT.

Lipolysis is a metabolic process that facilitates the release of stored energy from the adipose tissue and its transport to peripheral tissues to meet the body’s energy requirements during periods of increased demand. The imbalance between lipogenesis and lipolysis in lipid metabolism, which affects metabolic homeostasis, is considered a severe risk factor for metabolic disorders, including obesity [38]. Several vital genes, such as PPARGC1A, PPARα, ACOX1, and CPT1A, play roles in fatty acid transport, oxidation, and lipolysis. PPARGC1A helps to activate PPARα, whereas PPARα acts as a central regulator in fatty acid metabolism. ACOX1 and CPT1A play direct roles in fatty acid oxidation (β-oxidation) and are regulated by PPARα [39]. The role of PPARα in obesity is well-known; PPARα ligands are clinically used to treat obesity comorbidities such as dyslipidemia. For instance, certain citrus fruits or their extracts act as PPARα agonists that exhibit anti-obesity effects via PPARα activation and β-oxidation stimulation [40]. In this study, the CSHE treatment significantly elevated the mRNA levels of *Ppara* and *Acox1* in the liver of obese mice, indicating that CSHE promotes fatty acid β-oxidation and helps prevent lipid accumulation in the liver. Our previous report showed that the administration of *Palmaria mollis* red seaweed powder increased the *Ppara* and *Acox1* expression in the livers of diet-induced obese zebrafish and mouse models [41]. Moreover, the increased expression of *Ppargc1a* indicates that activated β-oxidation occurred in the eWAT of CSHE-treated obese mice. Overall, CSHE may suppress hepatic lipid accumulation via enhancing fatty acid β-oxidation and decreasing the volume of visceral adipose tissue by reducing lipogenesis in the eWAT. 

We further investigated the upstream factors and response of the signaling pathways to the CSHE treatment, including the protein expression and phosphorylation of the PI3K/AKT and AMPK signaling pathways (Figure 4, Figure 5, Figure 6, Figure 7 and Figure 8). For further clarification, we proposed a model to illustrate the possible mechanisms underlying the effects of CSHE on the upstream factors and signaling pathways involved in lipogenesis and lipolysis (Figure 9). The elevated phosphorylation of AKT and AMPK enhanced PPARα, which promotes lipolysis, and ACOX1, which transports fatty acids into the mitochondria via accelerated β-oxidation and suppressed lipid accumulation in the liver. In eWAT, increased the phosphorylation of FoxO1 leads to increased nuclear export [42,43], suppressed FASN expression, and decreased lipogenesis. Additionally, we hypothesized that the activated phosphorylation of ACC may lower malonyl-CoA, a substrate for fatty acid synthesis, causing irreversibly regulated enzyme reactions and increasing acetyl-CoA concentration, which can suppress the lipogenesis in eWAT [42]. To confirm this hypothesis, future studies are needed to analyze the changes in acetyl-CoA levels upon CSHE treatment and to correlate them with ACC phosphorylation levels.

PMFs are present in citrus fruits and suppress lipid accumulation via the PKA/AMPK/ACC signaling pathway [43]. Hence, the PMFs included in the CSHE treatment may induce similar signaling responses. Since CSHE is an extract derived from the peel and contains several components, including PMFs, it may influence two or more signaling pathways beyond those indicated in our proposed model, and their synergistic effects might emphasize the impact of CSHE on the suppression of lipid accumulation. In CSHE-treated 3T3-L1 adipocytes, high levels of phosphorylated ACC were detected, possibly contributing to decreased malonyl-CoA and lipid accumulation, consistent with observations in the HFD-induced obese mice after CSHE treatment.

Moreover, ectopic lipid accumulation, characterized by the accumulation of lipids in cells and tissues that are not prone to lipid storage, is well-documented and includes illustrative instances of fatty liver disease. This disease exhibits a signaling response similar to that associated with visceral and subcutaneous fat accumulation. The related mechanisms and contributing factors have been documented. Furthermore, the factors emerging in this context continue to be revealed [44,45]. According to this research, CSHE triggers the phosphorylation of AKT and AMPK in the liver, initiating a signaling cascade that leads to lipolysis. Nevertheless, it is plausible that various components of the CSHE may hinder the factors that facilitate ectopic lipid accumulation. Using a cell assay model, we confirmed that the CSHE treatment suppressed lipid droplet accumulation in HepG2 cells (unpublished data). Recently, the turmeric and honeyberry (Lonicera caerulea var. emphyllocalyx) extracts have been reported to suppress ectopic fat accumulation [46,47]. These results were in agreement with the suppressive action of CSHE on ectopic fat accumulation. Therefore, CSHE decreases visceral fat and suppresses ectopic lipid accumulation. Comprehensive knowledge of the functional components of the CSHE extract can reveal interesting facts that can support preventive medicine and pharmacology.

## 4. Materials and Methods

### 4.1. Ethics Statement

Animal experiments followed the Japanese Animal Welfare Regulatory Practice Act on the Welfare and Management of Animals (Ministry of the Environment of Japan) and international guidelines. All experimental procedures involving animals were approved by the Ethics Committee of Mie University, Tsu, Japan (Approval NO. 28-4).

### 4.2. Preparation of Hexane Extract from Citrus sphaerocarpa Peel

As previously reported, CSHE was obtained from the peels of *Citrus sphaerocarpa* [21]. Briefly, *n*-hexane was added to the peel and filtrated. Under reduced pressure, *n*-hexane was eliminated, and CSHE was obtained with a yield ratio of approximately 0.5%. The extracted samples were dissolved in dimethyl sulfoxide to prepare stock solutions (10–50 mg/mL). For the mouse experiments, a CSHE-lecithin emulsion was prepared using lecithin (PC70; Tsuji Oil Mills, Mie, Japan) to increase the water solubility of CSHE, as previously described [48]. PC70 indicates that more than 70% of the lecithin is composed of phosphatidylcholine. Two percent (*w*/*w*) of PC70 (20 mg/mL) was dissolved in 70% glycerol at 60 °C, and 1% (*w*/*w*) CSHE (10 mg/mL) was further added to the mixture, followed by complete dissolution at 60 °C. The mixture was first emulsified for five minutes at 60 °C with a disperser (Polytron Homogenizer PT2100, Central Scientific Commerce, Tokyo, Japan). The emulsions were treated twice at 100 MPa using an NM2-L200 emulsification machine (Yoshida Kikai, Aichi, Japan). The final emulsion, containing 2% lecithin and 1% CSHE, was used in this study.

### 4.3. Zebrafish Obesogenic Test (ZOT)

Zebrafish larvae were fed a Gemma Micro 75 or 150 diet, depending on their length (Skretting, Fontaine-lès-Vervins, France). Larvae with a standard 7–10 mm length were transferred to 6-well plates. Each well contained five larvae and 5 mL 0.3× Danieau’s solution (17.4 mM NaCl, 0.21 mM KCl, 0.12 mM MgSO_4_, 0.18 mM Ca(NO_3_)_2_, and 1.5 mM 4-(2-hydroxyethyl)-1-piperazinyl-ethane-2-sulfonic acid (HEPES) (pH 7.6)). On the first day of the short-term treatment (Figure 1a), larvae were fed with 0.1% hard-boiled chicken egg yolk as a high-fat diet; on the next day, larvae were stained with Nile Red working solution (5 μg/mL in 1% acetone-H_2_O) for 30 min at 28 °C in the dark and were washed twice using 0.3× Danieau’s solution for 5 min. The visceral adipose tissue was visualized and photographed using a BZ-X710 fluorescence microscope (TRITC filter; Keyence, Tokyo, Japan). Subsequently, the fish were divided into the HFD and CSHE treatment groups. The HFD control group was starved, and the CSHE group was exposed to 10 and 20 μg/mL of CSHE. After one day of treatment, Nile Red staining was performed again as described above. The fluorescence intensity was quantified using ImageJ software (Fiji distribution, version 1.52u, National Institutes of Health, Bethesda, MD, USA), and the intensity ratio (CSHE-treated group/HFD control group) was calculated. For long-term treatment, larvae were fed a HFD for one day and further starved for another day; subsequently, they were exposed to 10 μg/mL CSHE for 4 days. Nile Red staining was performed on days 1, 2, and 4 of the long-term treatment.

### 4.4. Mouse Experiments

C57BL/6NCrSlc (B6) male mice (8 weeks old; Japan SLC, Shizuoka, Japan) were divided randomly into four groups of five animals each: (1) ND group fed normal chow (CE-7; CLEA Japan, Tokyo, Japan) and drinking water containing 0.2% PC70 (lecithin; Tsuji Oil Mills, Mie, Japan); (2) ND + 0.1% CSHE group fed normal chow and drinking water containing 0.1% CSHE emulsion (containing a final concentration of 0.2% PC-70); (3) HFD group fed a HFD (60% of energy from fat; Test Diet 58Y1; TestDiet, Richmond, IN, USA) and drinking water containing 0.2% PC70; and (4) HFD + 0.1% CSHE group fed a HFD and drinking water containing 0.1% CSHE emulsion. During the six-week experiment, food and water intake were monitored twice per week, and body weights and FBG levels were measured weekly. At the endpoint, mice were euthanized via over-anesthesia using isoflurane (Pfizer, Pearl River, NY, USA), and the collected blood was used to measure their plasma TG and TCHO. A 3D-micro-CT was performed immediately after anesthesia using an in vivo System R_mCT 3D-micro-CT scanner (Rigaku, Tokyo, Japan). Three-dimensional images were reconstructed using i-View type R software (Version 1.67; J. Morita Mfg, Kyoto, Japan), and the volume of VAT was analyzed using CT Atlas Metabolic Analysis ver. 2.03 software (Rigaku, Tokyo, Japan). After the CT, the liver and eWAT were dissected for histological assays and qRT-PCR analysis.

### 4.5. Oil Red O Staining

Dissected mouse livers were fixed using a 4% formaldehyde solution dissolved in PBS (Histo-Fresh, Falma, Tokyo, Japan) at 4 °C for 24 h. Tissues were kept in 30 % sucrose solution for 2 h at room temperature and embedded in Tissue-Tek OCT compound (Sakura Finetek Japan, Tokyo, Japan), followed by rapid freezing in liquid nitrogen-cooled isopentane (Fujifilm Wako). Next, 8 μm sections of the tissues were prepared using the HM-550 cryostat (Microm, Walldorf, Germany). As previously described, liver sections were stained with 0.3% Oil Red O (Fujifilm Wako) [26]. Oil Red O quantification was performed using ImageJ software (Fiji distribution, version 1.52u).

### 4.6. Real-Time Quantitative Reverse Transcription PCR (qRT-PCR)

After bead homogenization of the tissues, total RNA was extracted from the mouse liver and eWAT using TRIzol reagent (Life Technologies, Carlsbad, CA, USA) [49]. The total RNA concentration was measured using a spectrophotometer (BioPhotometer, Eppendorf, Hamburg, Germany). cDNA was synthesized using 500 ng of total RNA and a ReverTra Ace qPCR RT Kit (Toyobo, Osaka, Japan). qRT-PCR analysis was performed using a Power SYBR Green Master Mix (Applied Biosystems, Foster City, CA, USA) and an ABI StepOnePlus Real-Time PCR System (Applied Biosystems). Relative mRNA expression levels were normalized using glyceraldehyde-3-phosphate dehydrogenase (GAPDH). The sequences, accession numbers, and amplicon sizes corresponding to the primers are given in Appendix A.

### 4.7. Histological and Immunofluorescence Staining In Vivo

For fixation, the dissected mouse liver and eWAT tissues were treated with 4% formaldehyde dissolved in PBS at 4 °C for 24 h. Tissues were dehydrated using graded ethanol and embedded in paraffin. Sections were cut to 3 μm thickness. After rehydrating, sections were placed in a pressure cooker filled with sodium citrate buffer to facilitate antigen retrieval. Sections were incubated with primary antibodies (Appendix A) overnight at 4 °C, followed by rinsing with Tris-buffered saline supplemented with 0.025% Triton X-100 (TBST). Next, the sections were incubated with a fluorescence-conjugated secondary antibody, Goat Anti-Rabbit IgG H&L (Alexa Fluor^®^ 488) preabsorbed (ab150081; Abcam, Cambridge, MA, USA) at room temperature (25 °C) for 1 h and mounted using ProLong Gold Antifade Mountant with DAPI (Thermo Fisher Scientific, Waltham, MA, USA). The fluorescence images were taken using a BZ-X710 microscope and quantified using ImageJ software (Fiji distribution, version 1.52u).

### 4.8. Cell Culture and CSHE Treatment

Mouse 3T3-L1 cells (JCRB Cell Bank, Osaka, Japan) were maintained in Dulbecco’s modified Eagle’s medium (DMEM; Fujifilm Wako) with 10% fetal bovine serum (FBS; Biowest, Nuaille, France) and 1% penicillin-streptomycin-l-glutamine (Fujifilm Wako) at 37 °C in a humidified 5% CO_2_ incubator. To stimulate adipocyte differentiation, 3T3-L1 cells were seeded in a 24-well plate (TPP, Trasadingen, Switzerland) with a cover glass placed at the bottom, cultured in an incubator until 100% confluence was achieved, and then maintained in a post-confluence state for 2 days (defined as day 0). Differentiation was induced by adding 10 µg/mL insulin (Sigma-Aldrich, St. Louis, MO, USA), 0.25 µM dexamethasone (Dex, Sigma-Aldrich), and 0.5 mM 3-isobutyl-1methylxanthine (IBMX, Sigma-Aldrich), with or without 50 µg/mL CSHE. After 2 days of incubation, the medium was replaced with DMEM supplemented with 10% FBS and 5 μg/mL insulin, with or without 50 µg/mL CSHE. The medium was renewed every two days over the course of 8 days.

### 4.9. In Vitro Immunofluorescence Staining and Image Quantification

After adipocyte differentiation, cells with or without CSHE were washed twice with 1× Tris-buffered saline (TBS) and fixed with 10% formalin solution (Fujifilm Wako) for 10 min at room temperature. After washing twice with 1× TBS, membrane permeabilization was performed for 10 min at room temperature using 0.1% Triton X-100 in 1× TBS, and the cells were washed with 1× TBS. Subsequently, cells were incubated with 3% bovine serum albumin (BSA) (Fujifilm Wako)/1× TBS for 1 h at 37 °C. ACC and pACC antibodies were diluted (1:200 and 1:400, respectively) with 1% BSA/1× TBS, dropped onto the cover glasses, and incubated overnight at 4 °C. After washing with TBS supplemented 0.05% Tween 20 (Nacalai Tesque, Kyoto, Japan) (TBST) three times, the cells were incubated with a secondary antibody, goat anti-rabbit 488 IgG (1:200; Thermo Fisher Scientific Waltham, MA, USA) solution diluted with 1% BSA/1× TBS, for 45 min at 37 °C. After washing three times with TBST, the nuclei were stained with Hoechst 33342 (1:10,000; Thermo Fisher Scientific Waltham, MA, USA) diluted in 1× TBS for 5 min at room temperature. After washing sequentially with 1× TBS and H_2_O, the cover glasses were covered with 90% glycerol/phosphate-buffered saline (PBS) supplemented with 2.5% diazabicyclooctane (Sigma-Aldrich) used as an antifade. The samples were examined using a fluorescence microscope (CKX53; Evident, Tokyo, Japan) and quantitatively analyzed using ImageJ software (Fiji distribution, version 1.52u).

### 4.10. Statistical Analysis

Based on the number of comparisons, statistical significance was determined via Student’s *t*-test or a one-way analysis of variance (ANOVA) with Bonferroni–Dunn multiple comparison tests using GraphPad Prism version 9.5.1 (GraphPad Software Inc., San Diego, CA, USA). Data were presented as mean ± standard deviation (SD). A *p*-value below 0.05 was considered significant.

## 5. Conclusions

CSHE supplementation reduces visceral adiposity in obese juvenile zebrafish. In HFD-induced obese mice, CSHE administration prevented an increase in body weight, suppressed the visceral adipose tissue’s volume, and alleviated lipid accumulation in the liver. Mechanistically, CSHE upregulated the expression of lipolysis-related genes in the liver and downregulated lipogenesis-related genes in the eWAT. Furthermore, the PI3K/AKT/FoxO1 and AMPK/ACC pathways, which are critical signaling pathways involved in lipid storage, were modulated by the CSHE treatment. These findings suggest that CSHE is a novel preventive and therapeutic agent for obesity. 

## Figures and Tables

**Figure 1 molecules-28-08026-f001:**
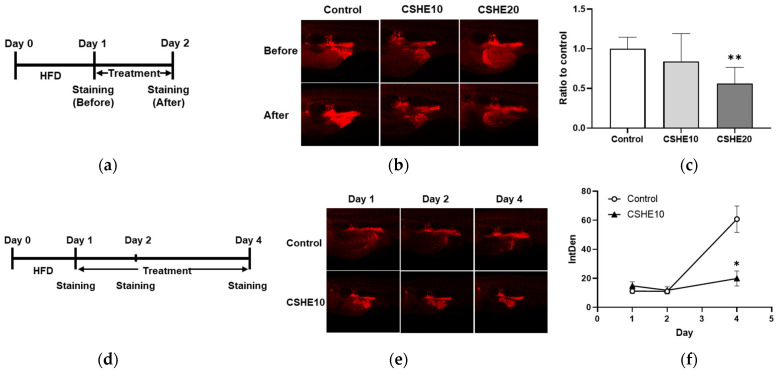
Hexane extract of *Citrus sphaerocarpa* (CSHE) suppressed the lipid accumulation in zebrafish juveniles. (**a**) The experimental design of the zebrafish obesogenic test (ZOT) after short-term treatment with CSHE. HFD, high-fat diet. (**b**) Nile Red (NR)-stained zebrafish juveniles were recorded 24 h before and after treatment with 10 μg/mL (CSHE10) and 20 μg/mL CSHE (CSHE20). The red color indicates NR-positive fluorescence detected in vascular adipose tissue (VAT). (**c**) Quantification of the fluorescence intensity in (**b**). (**d**) The experimental design of the ZOT after long-term treatment with CSHE. (**e**) Representative NR-stained VAT of the juveniles on days 1, 2, and 4 of their treatment with or without CSHE10. (**f**) Quantifying the intensity of NR-positive staining on days 1, 2, and 4. * *p* < 0.05, ** *p* < 0.01 vs. control, *n* = 5. Mean ± standard deviation (SD).

**Figure 2 molecules-28-08026-f002:**
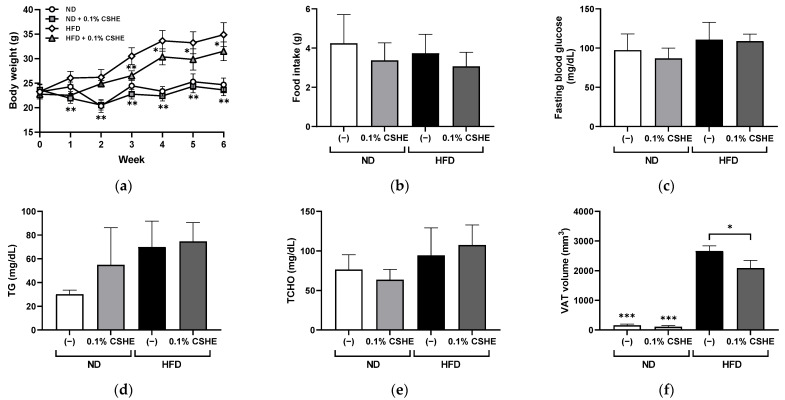
Effects of hexane extract of *Citrus sphaerocarpa* (CSHE)’s intake in obese mice. (**a**) Changes in the body weight over a six-week feeding period. (**b**–**f**) Changes in the food intake (**b**), fasting blood glucose (**c**), plasma triacylglycerol (TG) (**d**), plasma total cholesterol (TCHO) (**e**), and the visceral adipose tissue (VAT) volume (**f**) after the mice were fed a normal diet (ND) and high-fat diet (HFD) with or without CSHE treatment. (**g**) Representative 3D-micro-CT images. Red dotted lines delineate the region designated for measuring VAT volume, spanning from the xiphoid process of the sternum to the hip joint. The visceral and subcutaneous adipose tissues were indicated in yellow and orange, respectively. (**h**) Oil Red O staining analysis of liver sections. Red droplets show neutral lipid staining. (**i**) Quantification of the Oil Red O-positive areas. Scale bar = 50 μm. * *p* < 0.05, ** *p* < 0.01, *** *p* < 0.001 vs. HFD-treated control group, *n* = 5. Mean ± standard deviation (SD).

**Figure 3 molecules-28-08026-f003:**
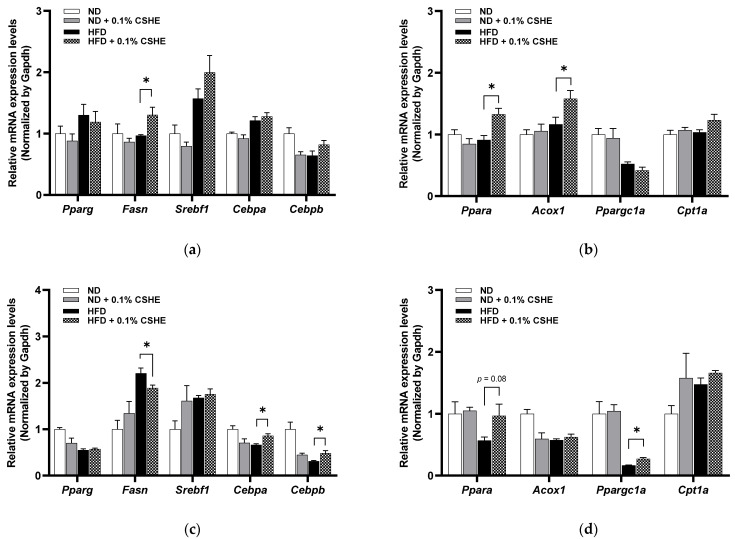
Expression levels of genes related to lipogenesis and lipolysis in the liver tissue and eWAT of mice treated with or without 0.1% CSHE. (**a**) Expression levels of lipogenesis-related genes in the liver. (**b**) Expression levels of lipolysis-related genes in the liver. (**c**) Expression levels of lipogenesis-related genes in eWAT. (**d**) Expression levels of lipolysis-related genes in the eWAT. ND, normal diet; HFD, high-fat diet; CSHE, hexane extract of *Citrus sphaerocarpa*; *Pparg*, peroxisome proliferator-activated receptor gamma; *Fasn*, fatty acid synthase; *Srebf1*, sterol regulatory element binding transcription factor 1; *Cebpa*, CCAAT/enhancer binding protein (C/EBP), alpha; *Cebpb*, CCAAT/enhancer binding protein (C/EBP), beta; *Ppara*, peroxisome proliferator-activated receptor alpha; *Acox1*, acyl-Coenzyme A oxidase 1, palmitoyl; *Ppargc1a*, peroxisome proliferative activated receptor, gamma, coactivator 1 alpha; *Cpt1a*, carnitine palmitoyltransferase 1a. * *p* < 0.05 vs. HFD-treated group, *n* = 5. Mean ± standard deviation (SD).

**Figure 4 molecules-28-08026-f004:**
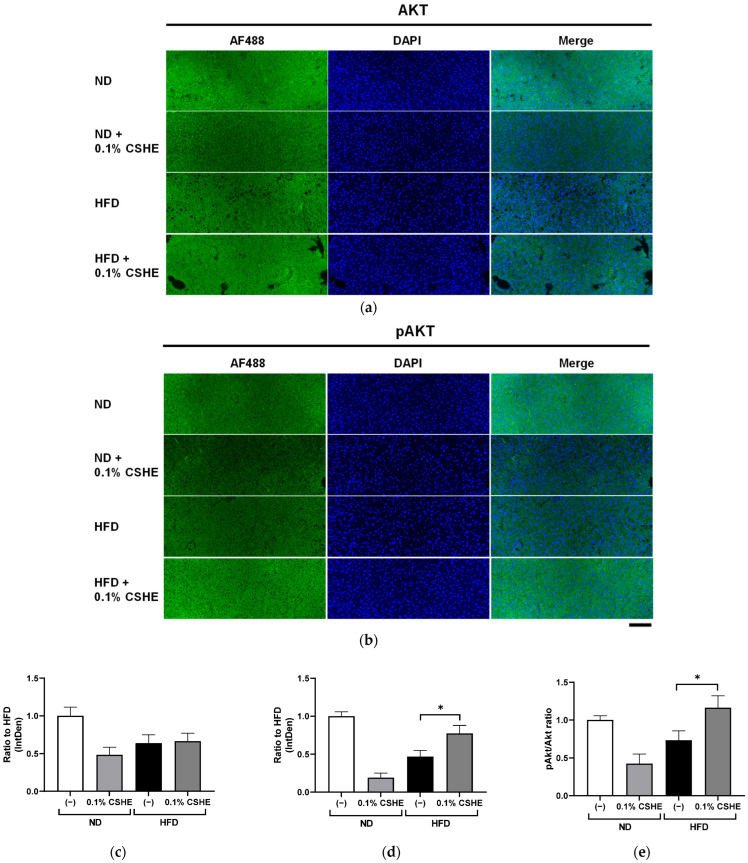
Immunofluorescence staining of AKT and phosphorylated AKT (pAKT) in liver tissues of mice treated with or without 0.1% CSHE. (**a**) Representative photographs showing AKT (green) expression in the liver tissues. (**b**) Representative photographs showing pAKT expression in the liver tissues. (**c**,**d**) AKT and pAKT protein levels were quantified using ImageJ; AKT activity is represented as a pAKT/AKT ratio (**e**). Nuclei were stained blue using the ProLong Gold Antifade Mountant with DAPI. Images were merged after quantification. Scale bar = 100 μm. ND, normal diet; HFD, high-fat diet. * *p* < 0.05 vs. HFD-treated control group; *n* = 12–15. Mean ± standard deviation (SD).

**Figure 5 molecules-28-08026-f005:**
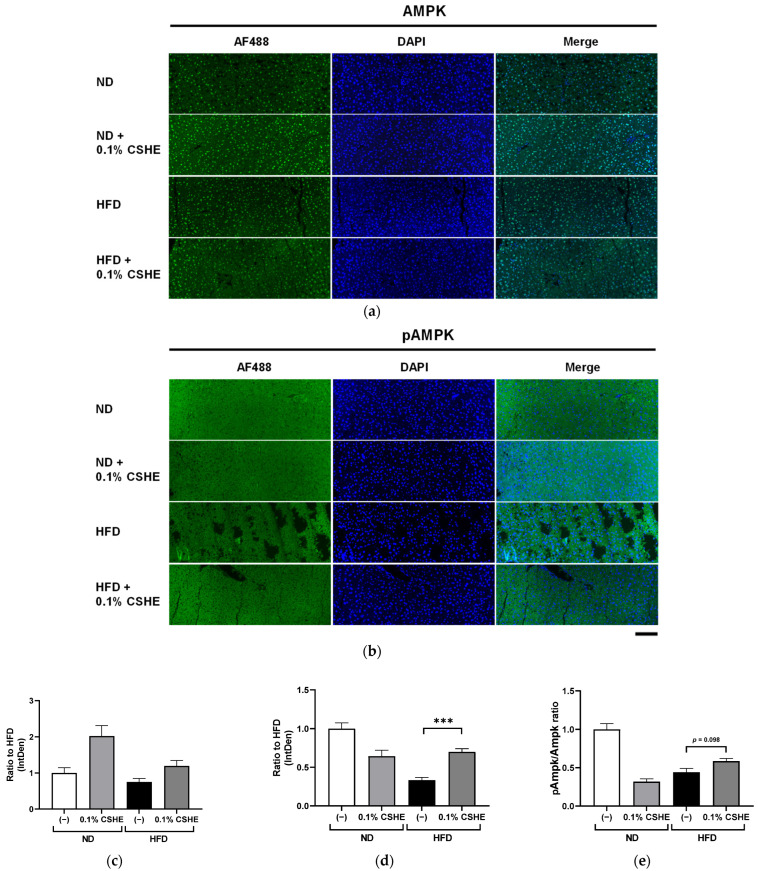
Immunofluorescence staining of AMPK and phosphorylated AMPK (pAMPK) in the liver tissue of mice treated with or without 0.1% CSHE. (**a**) Representative photographs showing AMPK (green) expressed in the liver tissue. (**b**) Representative photographs indicating pAMPK expression in the liver tissue. (**c**,**d**) AMPK and pAMPK protein levels were quantified using ImageJ, and the AMPK activity was quantified as a pAMPK/AMPK ratio (**e**). Nuclei were stained blue using the ProLong Gold Antifade Mountant with DAPI. Images were merged after quantification. Scale bar = 100 μm. ND, normal diet; HFD, high-fat diet. *** *p* < 0.001 vs. HFD-treated control group, *n* = 12–15. Mean ± standard deviation (SD).

**Figure 6 molecules-28-08026-f006:**
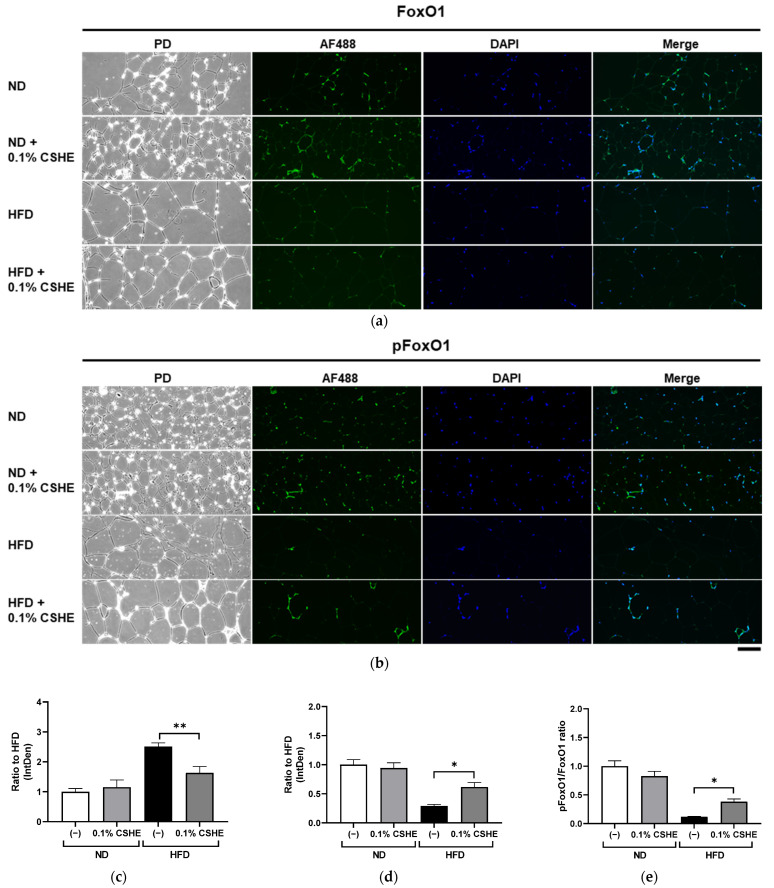
Immunofluorescence staining of FoxO1 and phosphorylated FoxO1 (pFoxO1) in the eWAT of mice treated with or without 0.1% CSHE. (**a**) Representative photographs showing FoxO1 (green) expression in eWAT. PD, phrase difference images. (**b**) Representative photographs showing pFoxO1 expression in eWAT. (**c**,**d**) Protein levels of FoxO1 and pFoxO1 were quantified using ImageJ; FoxO1 activity is represented as a pFoxO1/FoxO1 ratio (**e**). Nuclei were stained blue using the ProLong Gold Antifade Mountant with DAPI. Images were merged after quantification. The scale bar represents 100 μm. ND, normal diet; HFD, high-fat diet. * *p* < 0.05, **, *p* < 0.01 vs. HFD-treated control group, *n* = 12–15. Mean ± standard deviation (SD).

**Figure 7 molecules-28-08026-f007:**
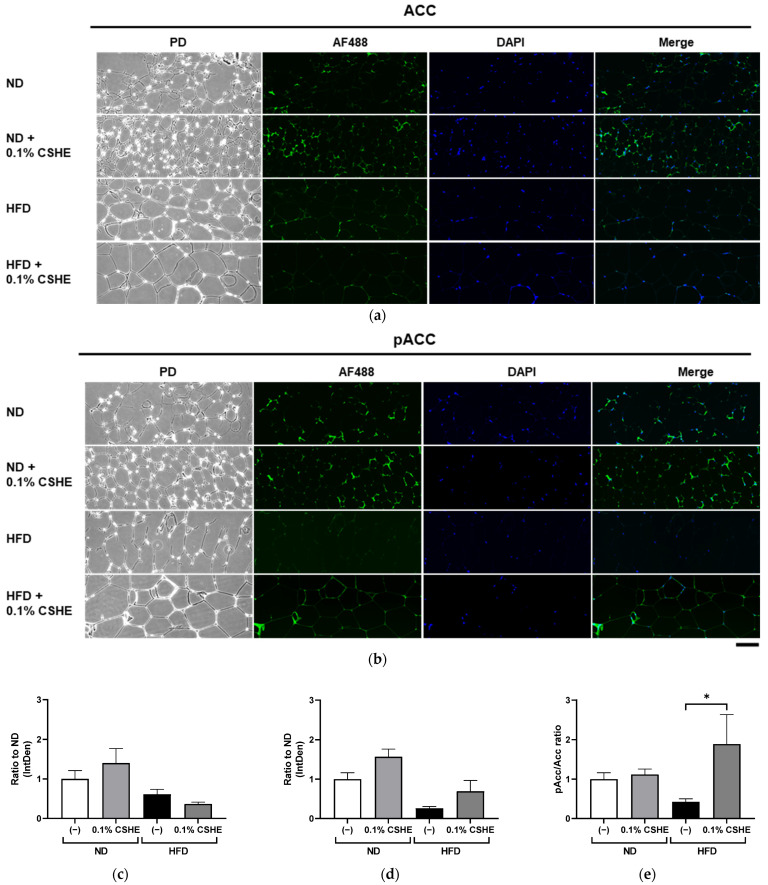
Immunofluorescence staining of the ACC and phosphorylated ACC (pACC) in the eWAT of mice treated with or without 0.1% CSHE. (**a**) Representative photographs indicating ACC (green) expression in eWAT. PD, phrase difference images. (**b**) Representative photographs showing the pACC expression in eWAT. (**c**,**d**) Protein levels of ACC and pACC were quantified using ImageJ; ACC activity is represented as a pACC/ACC ratio (**e**). Nuclei were stained blue using the ProLong Gold Antifade Mountant with DAPI. Images were merged after quantification. Scale bar = 100 μm. ND, normal diet; HFD, high-fat diet. * *p* < 0.05 vs. HFD-treated control group; *n* = 12–15. Mean ± standard deviation (SD).

**Figure 8 molecules-28-08026-f008:**
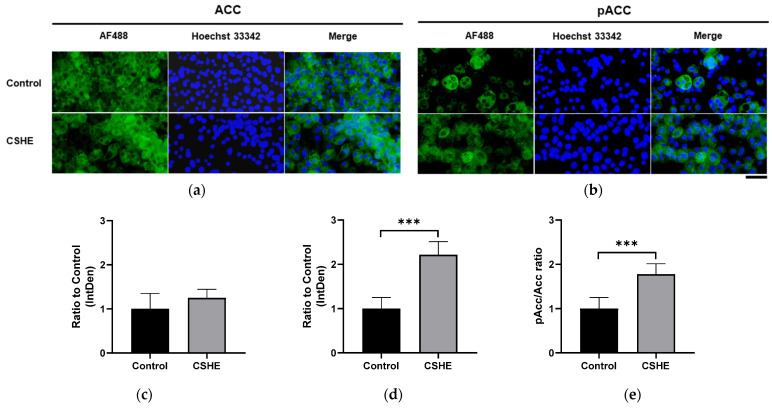
ACC expression and phosphorylation in differentiated 3T3-L1 adipocytes treated with 50 µg/mL CSHE. (**a**) Representative photographs showing ACC (green) expressions. (**b**) Representative photographs showing pACC (green) levels. (**c**–**e**) Levels of ACC and pACC were quantified using ImageJ; the pACC/ACC ratio was calculated. Nuclei were stained blue using Hoechst 33342. Hoechst 33342-positive fluorescence intensity was employed for normalization in the subsequent quantification and statistical analysis. Images were merged after quantification. Scale bar = 50 μm. *** *p* < 0.001 vs. control group, *n* = 10. Mean ± standard deviation (SD).

**Figure 9 molecules-28-08026-f009:**
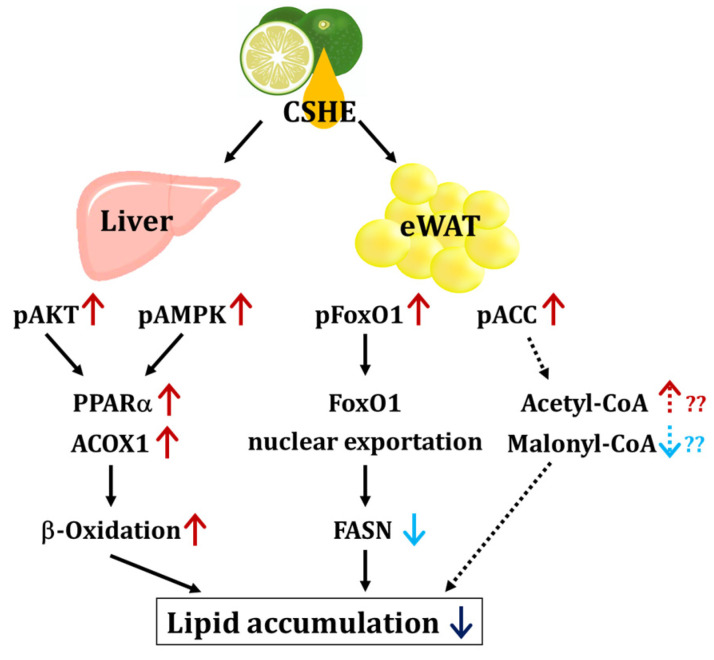
The proposed model of the signaling pathway induced by the CSHE treatment. CSHE: hexane extract of *Citrus sphaerocarpa*; eWAT: epididymal white adipose tissue; pAKT: phosphorylated protein kinase B; pAMPK: phosphorylated adenosine monophosphate-activated protein kinase; pFoxO1: phosphorylated forkhead box protein O1; pACC: phosphorylated acetyl-CoA carboxylase; PPARα: peroxisome proliferator-activated receptor alpha; ACOX1: acyl-coenzyme A oxidase 1 and palmitoyl; FoxO1: forkhead box protein O1; FASN: fatty acid synthase; Acetyl-CoA: acetyl coenzyme A; Malonyl-CoA: Malonyl coenzyme A. The red and blue arrows indicate the up- and down-regulation of protein and lipid metabolism, respectively. The dotted arrows represent the model estimated from the results obtained in this study.

## Data Availability

All of the relevant data are presented within the paper.

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
