# Peer review of "The Hexane Extract of Citrus sphaerocarpa Ameliorates Visceral Adiposity by Regulating the PI3K/AKT/FoxO1 and AMPK/ACC Signaling Pathways in High-Fat-Diet-Induced Obese Mice"

_molecules, 2023, doi:10.3390/molecules28248026_

Round 1

Reviewer 1 Report

Comments and Suggestions for Authors

The work done is very interesting. The methodology used is accurate, considering the background of research in this field. Regarding the presentation of the results, the authors could improve them, for example by eliminating the titles of each graph and improving the quality of the immunofluorescence photographs, mainly those of figures 6 and 7. Will it be possible to include abbreviations from figure 9? It is also necessary to review the abbreviations of the document. Regarding the discussion and conclusions, the arguments support the results obtained.

Review and compare lines: 535-536 Vs 545-546

Reviewer 2 Report

Comments and Suggestions for Authors

Dear respected editor;

-          Regarding the article number molecules-2738564; titled ‘‘Hexane Extract of Citrus sphaerocarpa Ameliorates Visceral Adiposity by Regulating PI3K/AKT/FoxO1 and AMPK/ACC Signaling Pathways in High-Fat Diet-Induced Obese Mice’’

- The similarity percentage is high (31%), needs to be minimized

-          I'm very grateful that you gave me the chance to review this essay.

-          The current study was designed to investigate the effects of CSHE on HFD-induced obesity in zebrafish and mice through evaluation of changes in genes and proteins involved in lipid metabolism in liver mouse and epididymal white adipose tissue (eWAT) to understand the underlying molecular mechanisms.

-          This study has a new idea as it uses important parameters and related genes and two different models to emphasis the findings.

-          The writing language is clear and understandable.

-          Despite, this study is one of the view studies concentrates on this issue, I still have some comments as follows.

Abstract:

-          The introduction could be minimized.

-          The objective needs to be revised correctly with more details.

-          You can add the CSHE doses used for treating zebrafish and mouse in methodology part.

-          Line 37-38; Moreover, CSHE is a promising novel preventive and therapeutic agent for managing obesity and associated diseases. Please correct it as you only study the obesity and we cannot generalized it on other disease.

Introduction:

-          Before the last paragraph, please add one sentence about the usage of zebrafish as a model in treating disease.

Materials and methods:

-          All abbreviations should be mentioned fully for the first time, then it could be abbreviated.

Results:

- In figure 2 e, please put better photos for histology as this one is not clear.         

In figures 2-4, please for all figure define all the abbreviations such ND and HFD

Discussion

 -          Please add small discussion and comparisons with other researches which study the same biomarkers and gens in obesity for both zebrafish or mouse and report the similarities and differences.

-          Conclusion: Good.

-          Funding: This research was funded by Tsuji Oil Mills Co. Ltd., please add the number of funding if there.

References;

-          Need to be corrected according to the journal instructions

-          All the references should be standardized.

Reviewer 3 Report

Comments and Suggestions for Authors

The manuscript describes the ameliorative effects of C. sphaerocarpa hexane extracts (CSHE) on visceral adiposity through PI3K/AKT/FoxO1 and AMPK/ACC signaling pathways in high-fat diet-induced obese mice. The authors analyze fat metabolism associated genes and performed immunostaining to confirm the signaling pathways. CSHE reduced fat accumulation in vascular adipose tissue and hepatocytes. Lipid metabolism-associated genes in liver (Pparα and Acox1) and epididymal white adipose tissue (Fasn) was regulated after CSHE treatment. This is an interesting study, and the results were attractive. However, in Fig. 9, the author summarized CSHE-induced variation of acetyl-CoA and malonyl-coA. There was no data on the analysis of these two metabolites in the study. The authors may perform the analysis to strength their claims.
